# 3, 3′- (3, 5-DCPBC) Down-Regulates Multiple Phosphokinase Dependent Signal Transduction Pathways in Malignant Melanoma Cells through Specific Diminution of EGFR^Y1086^ Phosphorylation

**DOI:** 10.3390/molecules27041172

**Published:** 2022-02-09

**Authors:** Abhijit Basu, M. Iqbal Choudhary, Karin Scharffetter-Kochanek

**Affiliations:** 1Department of Dermatology and Allergic Diseases, Ulm University, 89081 Ulm, Germany; Quratulain@iccs.edu (Q.-u.-A.); abhijit.basu@uni-ulm.de (A.B.); 2Dr. Panjwani Center for Molecular Medicine and Drug Research, International Center for Chemical and Biological Sciences, University of Karachi, Karachi 75270, Pakistan; iqbal.choudhary@iccs.edu; 3Department of Pathology, University of California, San Francisco, CA 94143, USA; 4H. E. J. Research Institute of Chemistry, International Center for Chemical and Biological Sciences, University of Karachi, Karachi 75270, Pakistan

**Keywords:** melanoma, *bis*-coumarin, tyrosine kinases, phosphorylation, phosphokinases, migration, proliferation, pathway analysis, ROS

## Abstract

Melanoma is the most dangerous skin malignancy due to its strong metastatic potential with high mortality. Activation of crucial signaling pathways enforcing melanoma progression depends on phosphorylation of distinct tyrosine kinases and oxidative stress. We here investigated the effect of a *bis*-coumarin derivative [3, 3′- ((3″, 5′-Dichlorophenyl) methylene) *bis* (4-hydroxy-2*H*-chromen-2-one)] [3, 3′- (3, 5-DCPBC)] on human melanoma cell survival, growth, proliferation, migration, intracellular redox state, and deciphered associated signaling pathways. This derivative is toxic for melanoma cells and non-toxic for melanocytes, their benign counterpart, and fibroblasts. 3, 3′- (3, 5-DCPBC) inhibits cell survival, migration, and proliferation of different metastatic and non-metastatic melanoma cell lines through profound suppression of the phosphorylation of Epidermal Growth Factor receptor (EGFR) and proto-oncogene cellular sarcoma (c-SRC) related downstream pathways. Thus, 3, 3′- (3, 5-DCPBC) endowed with the unique property to simultaneously suppress phosphorylation of multiple downstream kinases, such as EGFR/JAK/STAT and EGFR/SRC and their corresponding transcription factors.

## 1. Introduction

Malignant melanoma represents the most aggressive and deadliest form of skin cancer [1,2,3]. Several systemic therapies (cytotoxic chemotherapy, targeted drugs, immunotherapy, hormonal therapy, radiation therapy, and bio-chemotherapy) have been approved by the US Food and Drug Administration (FDA) [4,5]. Surgery still represents the first treatment option for primary melanoma where metastasis has not yet occurred. At later metastatic stages, systemic therapies are mandatory. The most successful treatment options against non-resettable metastatic melanoma are immunotherapies with antibodies directed against CTLA-4 and PD-1 [6,7] or their combination (small kinase and checkpoint inhibitors) [8,9,10]. The overall response rate of 10% to 15% for anti-CTLA-4 (ipilimumab), 25% to 45% for anti-PD-1 (nivolumab or pembrolizumab), and around 60% for their combination were embraced as a recent breakthrough in the therapy of a previously hard-to-treat malignancy [11]. However, due to the resistant nature of malignant melanoma [12], 40% of the patients do not respond to the combined therapy [13]. Due to severe and in part life-threatening side effects of immune checkpoint inhibitors affecting up to 50% of melanoma patients treated with anti-CTLA-4 immunotherapy [14], and even more when subjected to a combined anti-CTLA-4 and anti-PD-1 therapy [15], there is an urgent quest for new strategies in the battle against metastatic melanoma. Several promising agents have been tested against single protein kinases in malignancies [16,17]. 

Phosphorylated-receptors of tyrosine/serine/threonine protein kinases and their downstream targets are prime molecular players enforcing cell growth, survival, proliferation, and migration [18,19]. A number of these effectors within distinct signaling pathways are hyperphosphorylated and, consequently, hyperactivated in different cancers, including melanoma [20,21]. Some of the new therapeutic interventions against these kinases have successfully reached clinical trials, while others have already been approved by the US-FDA and entered clinical routine [22,23,24,25]. Currently, no single systemic therapy has successfully prevented melanoma progression in the long term. Due to the emergence of resistance against these drugs, multiple phase II and III melanoma trials are currently underway that are either studying the effect of combination treatments or striving for synthetic and natural compounds [26,27]. Combined therapies targeting two tyrosine kinases at least delay the development of resistance [28]. A newly emerging concept for treating advanced malignant melanoma is based on uncovering synthetic compounds targeting multiple signaling pathways and their corresponding genes. 

*Bis-*coumarins, a benzophenone family of natural compounds, have been reported to exhibit antioxidant, anti-inflammatory, anti-microbial, anti-glycation, anti-leukemic, and urease inhibition activities. They also have previously been studied as potential drug candidates against normal, non-cancerous, as well as various types of cancerous cancers cell lines [29]. They are, however, not endowed with the required potential to simultaneously target several signaling pathways in melanoma cells. We here set out to evaluate the effect of a *bis-*coumarins derivative, [3,3′-((3,5-dichlorophenyl)methylene) *bis* (4-hydroxy-2H-chromen-2-one)] [3, 3′- (3, 5-DCPBC)] on growth, survival, proliferation, and migration of non-metastatic and metastatic melanoma cells and to investigate its underlying molecular mode of action. In the present study, we demonstrated that treatment of human metastatic and non-metastatic melanoma cells with 3, 3′- (3, 5-DCPBC) profoundly diminished phosphorylation of the Epidermal Growth Factor Receptor (EGFR), proto-oncogene cellular sarcoma (c-SRC) and simultaneously suppressed phosphorylation of key downstream effectors and signaling pathways known to enforce melanoma progression. We also uncovered the non-toxic nature of this compound against human melanocytes, the benign counterpart of malignant melanoma cells. In aggregate, we here report for the first time a novel anti-melanoma drug candidate that not only suppresses cellular functions, key for melanoma progression but simultaneously targets multiple phosphor-tyrosine kinases. This is a clinically relevant advancement, and thus 3, 3′- (3, 5-DCPBC) may hold the unique therapeutic potential to be further tested in preclinical and clinical studies.

## 2. Results

### 2.1. 3, 3′-(3, 5-DCPBC) Induceds Cytotoxicity in Metastatic and Non-Metastatic Melanoma Cells 

3, 3′- (3, 5-DCPBC) was synthesized using two molecules of 4-hydroxy coumarin and 3, 5-dichlorbenzaldehyde in a condensation reaction [30,31], as shown in Figure 1a. Cytotoxicity of 3, 3′- (3, 5-DCPBC) was further assessed employing the MTT assay, where the activity of NADPH oxidoreductase served as a measure for the extent of cellular toxicity. The results were compared with the optical density determined for non-treated control cells (20% FBS DMEM) or dimethyl sulfoxide (DMSO). We found a concentration and time-dependent increase in cytotoxicity in A375 melanoma cells upon treatment with 3, 3′- (3, 5-DCPBC) (Figure 1b,c). While 90% of A375 melanoma cells were viable after incubation with 0.1 µM of 3, 3′- (3, 5-DCPBC) for 24 h, it depicts cytotoxicity of more than 50% of melanoma cells upon incubation at concentrations of 1, 10, and 100 µM for 24 h (Figure 1b). Cytotoxicity studies of 3, 3′- (3, 5-DCPBC) on WM-115 and metastatic SK-MEL-28 melanoma cells show similar results (Figure 1d,e).

To explore the specificity of 3, 3′- (3, 5-DCPBC) for its cytotoxicity on melanoma cells, the cytotoxicity of 3, 3′- (3, 5-DCPBC) was further assessed on human melanocytes (NHEM) and fibroblasts (FF95). Of note, 3, 3′- (3, 5-DCPBC) did not show any cytotoxicity on human melanocytes (NHEM) and fibroblast (FF95) at 12 and 24 h (Figure 1g,h).

### 2.2. 3, 3′- (3, 5-DCPBC) Inhibits the Migration of Melanoma Cells

Since cell migration is critical in cancer and melanoma progression [32,33], inhibition of melanoma cell migration is of prime importance. Multi-chamber Transwell^®^ migration assays were employed to assess the effect of 3, 3′- (3, 5-DCPBC) on random and directed migration of metastatic melanoma (A375, SK-MEL-28) and non-metastatic melanoma (WM-115) cells. The suppressive effect of 3, 3′- (3, 5-DCPBC) was first tested at three concentrations (0.1, 1, and 10 µM) for 4 h on the directed migration of A375 melanoma cells. Of note, 3, 3′- (3, 5-DCPBC) effectively suppressed the directed migration of A375 melanoma cells at all concentrations in a dose-dependent manner (Figure 2a). Type-IV collagen and 20% FBS served as strong chemoattractants (positive controls). 3, 3′- (3, 5-DCPBC) impressively inhibited directed A375 melanoma cell migration induced by 20% FBS (Figure 2a,b). Of note, 3, 3′- (3, 5-DCPBC) suppressed A375 melanoma cell migration by 93.1% at a concentration of 1 µM compared to the control (Figure 2a, middle panel). Representative photomicrographs of the bottom side of the perforated membranes confirm these results (Figure 2b). These data imply that 3, 3′- (3, 5-DCPBC) strongly suppresses A375 melanoma cell migration. To explore whether the migration suppressing efficacy of 3, 3′- (3, 5-DCPBC) is of general relevance, other metastatic and non-metastatic melanoma cells were assessed. Interestingly, the directed migration of the metastatic SK-MEL-28 melanoma cells and the non-metastatic WM-115 melanoma cells were similarly suppressed by 3, 3′- (3, 5-DCPBC) (Figure 2c). Figure 2d depicts the percentage of 3, 3′- (3, 5-DCPBC) suppression of directed melanoma cell migration. This strong inhibitory effect of 3, 5-DCPBC on directed melanoma cell migration is a true anti-migratory effect, not due to a 3, 3′- (3, 5-DCPBC) induced cytotoxicity (Figure 2e). Accordingly, more than 90% of A375 melanoma cells treated (0.1, 1, and 10 µM) concentrations of either with 3, 3′- (3, 5-DCPBC), or with DMSO (control) for 4 h Figure 2e were found to be viable as confirmed by MTT assay. In addition, 3, 3′- (3, 5-DCPBC) at a concentration of 0.1 µM did not show any cytotoxicity even after incubation of A375 melanoma cells for 24 h (Figure 1b). In contrast, there is a significant inhibition of A375 melanoma cell migration (Figure 2a). 

### 2.3. 3, 3′- (3, 5-DCPBC) Diminishes Metastatic Melanoma Cell Proliferation

The effect of 3, 3′- (3, 5-DCPBC) was further explored on the proliferation of metastatic A375 melanoma cells as a prime event in melanoma progression. For this purpose, BrdU incorporation was studied as previously described [34]. The DNA thymidine analog 5-Bromo-2′-deoxyuridine (BrdU) was incorporated into rapidly growing metastatic melanoma cells during the S phase of the cell cycle in the presence and absence of 3, 3′- (3, 5-DCPBC) and was detected fluorometrically by an antibody directed against BrdU at 548 and 576 nm excitation and emission, respectively. 3, 3′- (3, 5-DCPBC) efficiently suppressed melanoma cell proliferation at 48 h of treatment (Figure 3a). This anti-proliferative 3, 3′- (3, 5-DCPBC) effect corresponds to an average of the relative fluorescence of 1, compared to 4.9 of the DMSO control at the same time point (Figure 3a). 

In a complementary approach, BrdU positive cells from immunofluorescence photomicrographs were quantitated, and the percentage of BrdU positive cells were significantly reduced (Figure 3b). These data indicate a strong anti-proliferative effect of compound 3, 3′- (3, 5-DCPBC) on A375 metastatic melanoma cells.

### 2.4. 3, 3′- (3, 5-DCPBC) Does Not Alter the Intracellular Redox State

Metastatic melanoma progression is known to be driven by reactive oxygen species (ROS), in particular, migration and proliferation of metastatic melanoma cells are enhanced by an intracellular increase in superoxide (O_2_^−^) and hydrogen peroxide (H_2_O_2_) levels [35,36,37]. To investigate whether 3, 3′- (3, 5-DCPBC) exerts its anti-proliferative and anti-migratory effect by its intracellular free radical scavenging potential, the effect of 3, 3′- (3, 5-DCPBC) on mitochondrial and cytosolic superoxide (O_2_^−^), and H_2_O_2_ levels were studied employing the DHE and MitoSOX assays [38]. DHE can detect cytoplasmic O_2_^−^ and H_2_O_2_, MitoSOXTM Red identifies mitochondrial O_2_^−^. Briefly, A375 melanoma cells were incubated with MitoSOXTM Red or DHE dye in the presence and absence of 3, 3′- (3, 5-DCPBC), and fluorescence was measured by a fluorescence spectrophotometer. High fluorescence indicates a low free radical scavenging potential of 3, 3′- (3, 5-DCPBC). 3, 3′- (3, 5-DCPBC) at all tested concentrations did not exhibit any significant antioxidant effect on cytosolic or mitochondrial O_2_^−^ and H_2_O_2_ in A 375 melanoma cells (Figure 3d,e). By contrast with 3, 3′- (3, 5-DCPBC), both rotenone, a complex I inhibiting agent which H_2_O_2_ increases mitochondrial O_2_^−^, and *N*-acetyl cysteine (NAC) serving as a substrate for quenching, depict strong effects on ROS levels (Figure 3d–g). We found that 3, 3′- (3, 5-DCPBC) did not modulate intracellular redox state by scavenging mitochondrial/cytosolic O_2_^−^ or H_2_O_2_ concentrations. These data imply that the observed suppression of proliferation and migration on melanoma cells is not due to the antioxidant properties of 3, 3′- (3, 5-DCPBC). 

### 2.5. Molecular Mechanism of Action of 3, 3′- (3, 5-DCPBC)

A phospho-proteome profiling array [39] was employed to gain insight into the mechanism for 3, 3′- (3, 5-DCPBC) inhibiting melanoma cell survival, growth, proliferation, and migration; 1 × 10^5^ A375 melanoma cells were treated with 3, 3′- (3, 5-DCPBC) at a concentration of 1 µM for 4 and 18 h. For each time point, a total of 43 kinase phosphorylation sites and 2 related proteins (Heat Shock Protein and Tumor Suppressor Protein) were analyzed. A significant effect of 3, 3′- (3, 5-DCPBC) was observed on 31 phosphorylation sites of 25 different kinases at 4 h and 11 phosphorylation sites of six other kinases at 18 h. These data suggest that 3, 3′- (3, 5-DCPBC) targets multiple kinases (Figure 4a–d).

A list of non-affected kinases are shown in Appendix A. Densitometric analysis of significant phosphorylation sites of 25 kinases and their associated *p* values is shown in Figure 4e. Sixteen different kinases were more than 2-fold down-regulated upon treatment with 3, 3′- (3, 5-DCPBC) compared to DMSO as represented by the blue color (Figure 4f). The MA plot depicts Log2 fold changes on the *y*-axis, and Log2 mean expression on the *x*-axis (Figure 4f). Eleven kinases did not show any significant changes. Among the down-regulated phosphokinases, predominantly members of the family of the tyrosine kinase family were suppressed, phosphorylation of serine or threonine kinases was less frequently affected (Figure 4g). The *y*-axis represents the fold change, and the *x*-axis represents the position of the phospho-sites in the gene amino acid sequence (Figure 4g). Of all the kinases significantly downregulated by 3, 3′- (3, 5-DCPBC), the tyrosine kinase phospho-motif (blue filled circles) is the most affected phospho-motif in A375 melanoma cells treated with 3, 3′- (3, 5-DCPBC) as compared to threonine (green filled circles) or serine (pink filled circles). Phospho-motif mapping thus allowed us to uncover major kinases and their key phosphorylation sites that likely are involved in uncontrolled signaling in melanoma progression. Pathway analysis was further employed for the most suppressed kinases with downregulation of phospho-motifs to explore which signaling pathways are most prominently affected (Figure 5a). The *x*-axis represents the gene ratio, which is presented as % of total differential gene expression (DGE) in all GO clusters, and the *y*-axis represents the associated pathway. The color refers to the *p*-value and the count as the number of occurrences of this gene per GO cluster. Pathway analysis with the differentially down-regulated phosphor-motifs depicts receptor tyrosine kinases, inflammatory interleukins, ERK1/2, AKT, and mTOR as the major signaling cascades that are affected after treatment of melanoma cells with 3, 3′- (3, 5-DCPBC) for a period of 4 h. Briefly, phosphor tyrosine modulation at a specific residue of Epidermal Growth Factor Receptor (EGFR^Y1086^) and Fc-gamma Receptors^Y412^, and their downstream targets JAK/STAT were the most suppressed targets, including STAT2^Y689^, STAT5α^Y697^, STAT5α/β^Y694/Y699^, and STAT6^Y641^. Among the other 3, 3′- (3, 5-DCPBC) suppressed pathways, mTORS^2448^, ERK1/2^Y204/Y187^, SRC^Y419^, and β-Catenin^Y654^ were identified as highly important for melanoma progression. To understand whether inhibition of these phosphorylation sites following treatment of A375 melanoma cells with 3, 3′- (3, 5-DCPBC) will persist, phosphokinase array analysis was performed from lysates of A375 melanoma cells, which had been treated with 3, 3′- (3, 5-DCPBC) for 18 h (Figure 5b). The normalization of phosphoproteins expression by the β actin expression for 4 and 18 h depicts that downregulation of phosphorylation sites as observed at 4 h persists in many previously identified phosphokinases at 18 h. 

A total of 12 phosphorylation sites of nine different phosphoprotein kinases were found to be persistently downregulated both at 4 and 18 h (Figure 5b,c). These data imply that they constitute likely targets for 3, 3′- (3, 5-DCPBC) 

Treatment of A375 melanoma cells with 3, 5 DCPBC for 4 and 18 h profoundly suppressed phosphorylation of critical targets like EGFR, SRC, STAT3, JNK1/2, and MSK. Mainly phosphorylation sites of tyrosine kinases were suppressed (Figure 4g and Figure 5c) and, consequently, attenuated those target effectors downstream of tyrosine kinase signaling, which enforces melanoma progression.

## 3. Discussion

The significant unprecedented finding of this study is that the *bis*-coumarin derivative 3, 3′- (3, 5-DCPBC) has profound inhibitory properties on critical steps of malignant melanoma progression. Accordingly, proliferation, migration, and survival of melanoma cells—by contrast to benign melanocytes and fibroblasts—are impressively downregulated in-vitro in the presence of moderate concentrations of 3, 3′- (3, 5-DCPBC). 

Although, enhanced ROS concentrations have been reported to be associated with melanoma progression [35,36,37]. However, we found that the strong effect of 3, 3′- (3, 5-DCPBC) on critical features of melanoma progression is not due to antioxidant properties but is instead a consequence of its outstanding suppressive potential of different important tyrosine phosphokinases involved in melanoma progression (Figure 6). Phosphokinases modulate several cellular functions [40], and activation of multiple phosphokinases enforces the progression of melanoma and many other malignancies [41]. In particular, autophosphorylation of EGFR with phosphorylation of the downstream pathways play prime roles in melanoma progression [42]. Of note, 3, 3′- (3, 5-DCPBC) significantly diminished the phosphorylation of the Y1086 of EGFR at 4 and to a lesser extent at 18 h (Figure 4e,f and Figure 5b,c and Appendix A). 

This is most interesting as phosphorylation of this EGFR residue regulates various signal transduction pathways (mTOR, SFK, JAK-STAT, MSK, ERK) involved in cell proliferation, cell migration, and cell survival. Notably, these downstream pathways are hyperphosphorylated in various cancers, including melanomas [43,44,45,46,47]. In addition, hyperphosphorylation of mTOR at its threonine Thr2446 and serine residues (Ser2448 and Ser2481) occurred both via the EGFR-ERK-S6K1 axis and the PI3K/AKT axis [48] and is related to growth in various types of cancers and melanoma [49]. We further explored downstream pathways and the associated genes that are regulated through EGFR^Y1086^ phosphorylation (Figure 5a). Of note, EGFR/JAK/STAT and EGFR/SRC tyrosine kinases, the most important among them EGFR and SRC, and their downstream effectors were markedly downregulated upon 3, 3′- (3, 5-DCPBC) treatment as confirmed by downregulation of 16 genes coding for distinct phosphotyrosine kinases (Figure 4f). Suppression of phosphorylation (activation) of the mTOR pathway (mTOR, PRAS40, and ERK1/2/3) was observed mainly after 4 h of 3, 3′- (3, 5-DCPBC) treatment. EGFR^Y1086^ phosphorylation can activate the SRC family of kinases (SFKs) [50]. Tyrosine phosphorylation of members of SKFs plays a key role in cell differentiation, motility, proliferation, and survival [50,51,52]. Remarkably, we found profound suppression of phosphorylation of SKF members, among them SRC^Y419^, LYN^Y397^, LCK^Y394^, FYN^Y420^, YES^Y426^, FGR^Y412^, HCK^Y41^, and FAK^Y319^ in A375 malignant melanoma cells after treatment with 3, 3′- (3, 5-DCPBC). 

Our data on the importance of 3, 3′- (3, 5-DCPBC) induced suppression of EGFR ^Y1086^ phosphorylation is further underscored by the recent finding that growth hormone via EGFR and SRC kinases upregulates the transcription factor c-Kit for melanogenesis, and more importantly, that mutated Kit enforces MITF (Microphthalmia-associated transcription factor)-dependent transcription programs in melanoma [53,54]. Finally, recent data on the prime role of the tyrosine kinases SRC and the AKT/mTOR axis in melanoma progression [55,56] highlights the unique implication of 3, 3′- (3, 5-DCPBC) as a promising small molecule drug candidate. The finding that 3, 3′- (3, 5-DCPBC) is endowed with the unique property to simultaneously suppress many phospho-tyrosine kinases further underscores its potential prime clinical relevance. Interestingly, simultaneous inhibition of tyrosine phosphorylation of EGFR and SRC kinases can overcome the frequently developing BRAF resistance in melanoma [57,58]. Finally, phosphorylation on Y705/727 and Y694/699 of STAT3 and STAT5α/β most likely in cooperation with MSK enforce phosphorylation of ERK1/2 (extracellular signal-regulated kinase) or p38 in a growth factor receptor-independent signaling pathways [59]. These pathways are essential for melanoma progression, and their inhibition by 3, 3′- (3, 5-DCPBC) likely suppresses melanoma progression. 

## 4. Materials and Methods

### 4.1. Cell Lines, Reagents, and Antibodies

Metastatic melanoma cell lines; A375 (#CRL-1619), SK-Mel-28 (#HTB-72), WM-266-4 (#CRL-1676), and non-metastatic melanoma cell lines; WM115 (#CRL-1675) were procured from A.T.C.C, and normal fibroblasts (FF95) were provided by the Department of Dermatology, University of Ulm, Ulm, Germany. Human epidermal melanocytes (NHEM, #C-12400) (M2 media, #C-24300) were purchased from Promocell GmbH, Heidelberg, Germany. Thiazolyl blue tetrazolium bromide (MTT) dye, #M655, Accutase solution, (#A6964), Human placental type-IV collagen (#C5533), and DMSO (#276855) were purchased from Sigma Aldrich GmbH. Diff Quick staining solution kit (#B4132-1A) was purchased from Medion Diagnostic AG, Bern, Switzerland. Primary antibodies: anti-BrdU antibody #555627 was purchased from BD Biosciences, Waltham, MA, USA. Secondary antibody (Alexa fluor 555 goat anti-mouse, (#A32727) and BrdU dye (#B23151) were obtained from Thermofisher Scientific GmbH, Bremen, Germany. Human phospho-kinase array kit #ARY003B was purchased from R&D System, Minneapolis, MN, USA. Transwell^®^ chambers, #3422 was procured from Corning, NY, USA and Chemiluminescence reagent (Signal Fire, #12630S, Cell Signal) was purchased from Cell Signaling Technology, Danvers, MA, USA. Chamber slides (Millicell EZ, #C86024 Millipore/Merck GmbH) were purchased from Corning, NY, USA. Studied compound [3,3′-((3,5-chlorophenyl) methylene) *bis* (4-hydroxychroman-2-one)] [3, 3′- (3, 5-DCPBC)] was obtained from the in-house Molecular Bank of Dr. Panjwani Center for Molecular Medicine and Drug Research, International Center for Chemical and Biological Sciences, University of Karachi, Karachi, Pakistan.

### 4.2. Cell Culture

Metastatic melanoma cell lines; A375, SK-Mel-28, non-metastatic melanoma cell line; WM-266-4, WM115, and FF95 fibroblasts were grown in DMEM supplemented with Pen/Strep and 20% FBS at 37 °C, and 5% CO_2_. Cells were synchronized with starving medium (0.2% F-10 Nut Mix Ham 1X) for a further 12 h. After that, cells were either incubated with 3, 3′- (3, 5-DCPBC), or w/o in a control group, incubated with identical volumes of DMSO in DMEM for the indicated concentrations and incubation times. Human primary melanocytes were grown in M2 media.

### 4.3. Cell Cytotoxicity Assay

The MTT assay was employed to evaluate cytotoxicity as previously described [60]. Melanoma cells and their benign control cells were grown in 96 well plates, as described in Section 2.2. MTT solution was added for 4 h, and formazan crystals were dissolved in DMSO for 15 min at room temperature. Absorbance was recorded at 550 nm, and the reference wavelength was set to 650 nm using a microplate reader (Varioskan Lux, Thermofisher Scientific). Cytotoxicity of 3, 3′- (3, 5-DCPBC) was calculated as the relative ratio of optical densities compared to non-treated controls.

### 4.4. Transwell Migration Assay

The Transwell^®^ migration assay was performed as earlier described, with few modifications [61,62,63]. In brief, 1 × 10^5^ cells in 200 μL of either starved medium or 20% FBS in DMEM were loaded onto 8-micrometer pore Transwell inserts (upper chambers) and incubated for 2 h at 37 °C in 5% CO_2_. The lower chambers were loaded either with 600 µL of 20% FBS in DMEM, chemotactic stimuli (human placental type IV collagen) at a concentration of 100 μg/µL or dissolved in 20% FBS in DMEM, and incubated for 2 h at 37 °C in 5% CO_2_. Thereafter, cells were fixed and stained using the Diff Quik^®^ Stain kit. Non-migrated cells were removed from the upper chambers using cotton swabs. Perforated filters were removed and fixed on microscopic slides. An average number of migrated cells was counted in five high-power microscopic fields (HPF), randomly chosen at 200X magnification from each of the three technical and four biological replicates. Images of cells that migrated to the downside of the perforated membrane were collected using a Nikon TE300 inverted epifluorescence microscope.

### 4.5. Trypan Blue Cell Viability Assay

Melanoma and control cells were incubated with different concentrations of 3, 3′- (3, 5-DCPBC) for 4 h. After that, viable cells were counted using Vi-CEL XR 2.03 (Beckman Coulter, Brea, CA, USA) automated cell counter, as described earlier [64].

### 4.6. Bromodeoxyuridine (BrdU) Incorporation Assay

Melanoma and control cells were seeded in 96-well plates as described in Section 2.2, followed by the treatment of 1 µM 3, 3′- (3, 5-DCPBC)/100 µL of 20% FBS in DMEM in each well for different incubation times. After incubation, 3 µg/µL BrdU was added to each well for another 2 h at 37 °C, followed by fixation of cells with 1:1 acetone solution for 30 min. DNA strains were denatured with 2N HCl, and non-specific binding sites were blocked by 5% BSA dissolved in TBST. 50 µL primary anti-BrdU antibody was suspended in TBST at a dilution of 1:10 and added to detect the incorporated BrdU dye. Melanoma cells and control cells in each well were then treated for 2 h with 50 µL Alexa Fluor conjugated 555 goat-anti-mouse secondary antibodies diluted in TBST at 1:200 dilution. After washing melanoma and control cells with TBST buffer, fluorescence in each well containing 100 µL TBST was measured at 548 nm excitation and 576 nm emission.

### 4.7. BrdU Immunofluorescence Staining

Cells were grown in Millicell EZ chamber slides coated with poly-l-lysine and were further incubated with 100 µM BrdU for 2 h. Melanoma cells and control cells were fixed with 1:1 acetone solution for 30 min, DNA was hydrolyzed with 2N HCl for 30 min, and non-specific binding sites were blocked by 5% BSA dissolved in TBST. Melanoma cells and control cells were incubated overnight with 500 µL primary anti-BrdU antibody in TBST at a 1:10 dilution, washed thrice with TBST buffer. After that, cells were treated with 500 µL, Alexa Fluor conjugated 555 goat anti-mouse secondary antibody suspended in TBST at a 1:200 dilution in each well for 2 h at room temperature. Cell nuclei were stained with DAPI (1 µL/mL of DAPI in PBS) for 10 min, cells were fixed with formaldehyde and mounted with Fluoromount (Dako). Melanoma and control cells were examined under a fluorescence microscope at 40× magnification with AxioVision A5 microscope (Zeiss Inc., Oberkochen, Germany). BrdU-positive melanoma and control cells were counted manually from three independent images. The percentage of proliferating cells was assessed by calculating the number of BrdU-positive cells in the 3, 3′- (3, 5-DCPBC) treated group, and the DMSO treated control group in 5 to 6 random fields.

### 4.8. Messurement of Intracellular Radical Scavenging Activity

Intracellular cytosolic and mitochondrial superoxide radical scavenging activity was measured through fluorimeter using MitoSOXTM Red and DHE dyes. Briefly, 105 A375 cells grown in DMEM media were further starved in F-10 Nut Mix (Ham) for a period of 12 h, thereafter, media was aspirated, and 100 µL of sample solution (1 µM of each test compound dissolved DMEM) was added in each well of the 96-well plate. After 12 h. incubation cells were washed thrice with PBS and treated with 2.5 µM MitoSOXTM Red and 5 µM DHE dyes for 15 and 20 min at 37° respectively. After incubation, the cells were rinsed with PBS and to each well was added 100 µL buffer (PBS with freshly added 5 mm glucose). Finally, DHE fluorescence was measured at 510 nm excitation, and 595 nm emission, and MitoSOXTM Red fluorescence was measured at 510 nm excitation, and 595 nm emission. In both assays rotenone and N-acetylcysteine and rotenone were used as positive and negative controls respectively. However, DMSO was used as vehicle control.

### 4.9. The Human Phospho-Kinase Array Assay and Western Blot Analysis

The phosphorylation level of kinases was determined with the Proteome Profiler Array Kit (R&D Systems) according to the manufacturer’s instructions. Protein concentrations were determined by the Bradford protein assay. To block non-specific sites, each membrane was incubated in an array blocking buffer for 1 h. 3, 3′- (3, 5-DCPBC), and DMSO treated cell lysates (334 µL cell lysate/1 mL of array buffer corresponding to 200 µg protein lysate) were applied on membranes and incubated overnight. Thereafter, membranes were washed with 1X washing buffer followed by incubation with 20 mL of the detection antibody for 2 h on a shaker at room temperature. Membranes were thoroughly rinsed with washing buffer thrice and further incubated with Streptavidin-HRP for 30 min at room temperature. Membranes were washed with 1× washing buffer for 10 min, and after that, all membranes were simultaneously exposed to SignalFire plus chemiluminescent reagents for 1 min. Phospho-kinase array data were developed on Vilber FusionFx Chemiluminescence Imager for 1 to 10 min with multiple exposure times.

### 4.10. Statistical Analysis

Graph Pad Prism software 5 (GraphPad Software, Inc., San Diego, CA, USA) was used to analyze data. Parametric one-way analysis of variance was used with Tukey’s posthoc analysis to compare multiple groups of 3, 3′- (3, 5-DCPBC) treated, and DMSO treated groups. Two-way analysis of variance was used with Tukey’s posthoc test to compare two different time points and groups of 3, 3′- (3, 5-DCPBC) treated, and DMSO treated groups. The Student’s *t*-test (two-tailed) was employed to compare the two groups of counted cells taken from representative photomicrographs. Significance was defined as *p* < 0.05. *p* values were assigned * with *p <* 0.01, ** with *p <* 0.001 and *** with *p <* 0.0001. R studio version 1.1.463 was used for data analysis and networking analysis, respectively.

## 5. Conclusions

In conclusion, with the urgent need for new therapeutics to target multiple tyrosine kinases as an emerging concept for advanced melanoma, 3, 3′- (3, 5-DCPBC) might hold promise for an efficient alternative approach to currently established therapies [49,57].

We discovered that 3, 3′- (3, 5-DCPBC) is highly suppressive on many steps and pathways of melanoma cell progression at very low concentrations. At the same time, it is non-toxic for non-tumorous melanocytes and fibroblasts. The efficient combined targeting of EGFR, SRC, STAT, and MAPK by 3, 3′- (3, 5-DCPBC) in melanoma cells may long term assist clinicians to prevent melanoma growth and even to overcome drug resistance. Further preclinical data are now required to use this compound in future clinical trials.

## Figures and Tables

**Figure 1 molecules-27-01172-f001:**
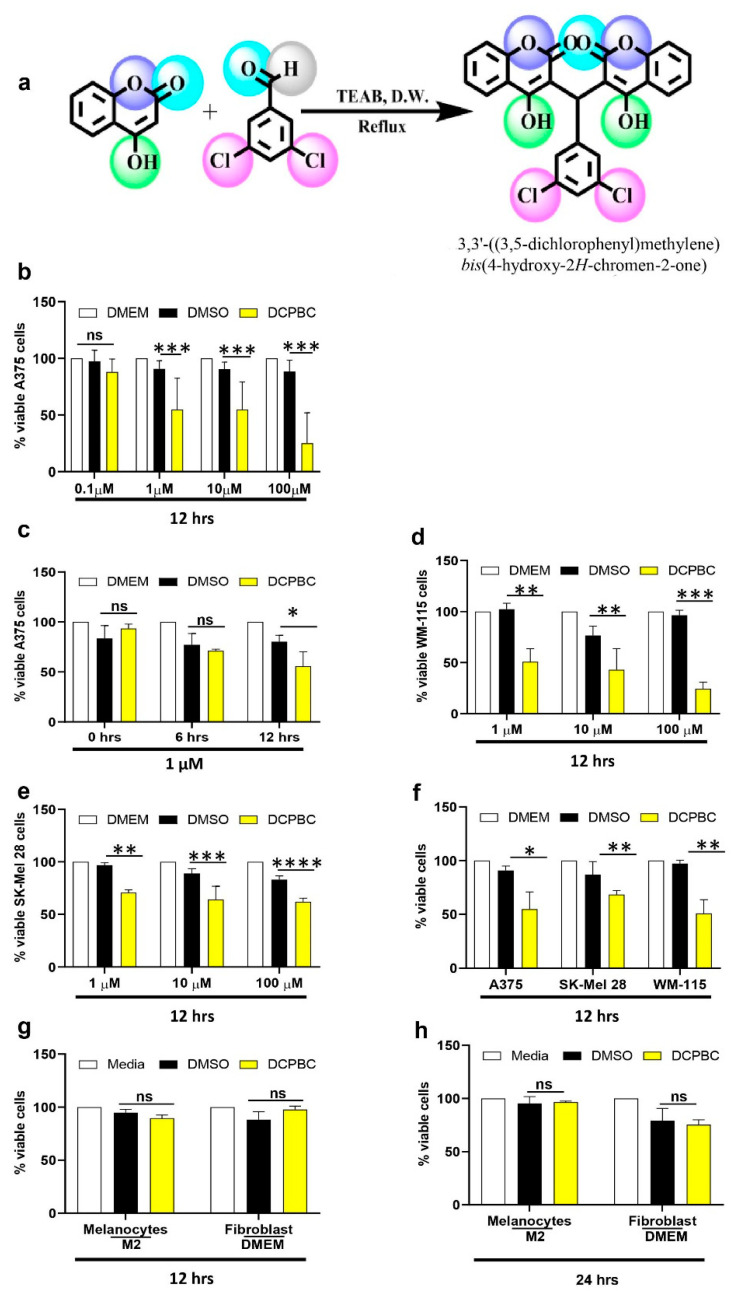
Synthesis of 3, 3′- (3, 5-DCPBC) and its cytotoxic effects on different melanoma cells. (**a**) Scheme depicting the synthesis of 3, 3′- (3, 5-DCPBC) using 3, 5-dichlrobenzaldehyde with 4-hydroxy coumarin in the presence of tetra ethyl ammonium bromide (TEAB). (**b**–**h**) Cytotoxic effect of 3, 3′- (3, 5-DCPBC) on metastatic melanoma cells, non-metastatic melanoma cells, primary melanocytes, the benign counterpart of melanoma cells, and fibroblasts at the indicated concentrations (0.1, 1 µM and 10 µM) and time point (12 and 24 h). M2 medium for melanocytes and DMEM for fibroblasts served as controls (**g**,**h**). An equal concentration of DMSO dissolved in growth media for melanocytes (M2) and fibroblasts (DMEM) was used as DMSO controls (**g**,**h**). Data are presented as a percentage of control (*n* = 3) ± S. D. DMSO, Dimethyl sulfoxide.

**Figure 2 molecules-27-01172-f002:**
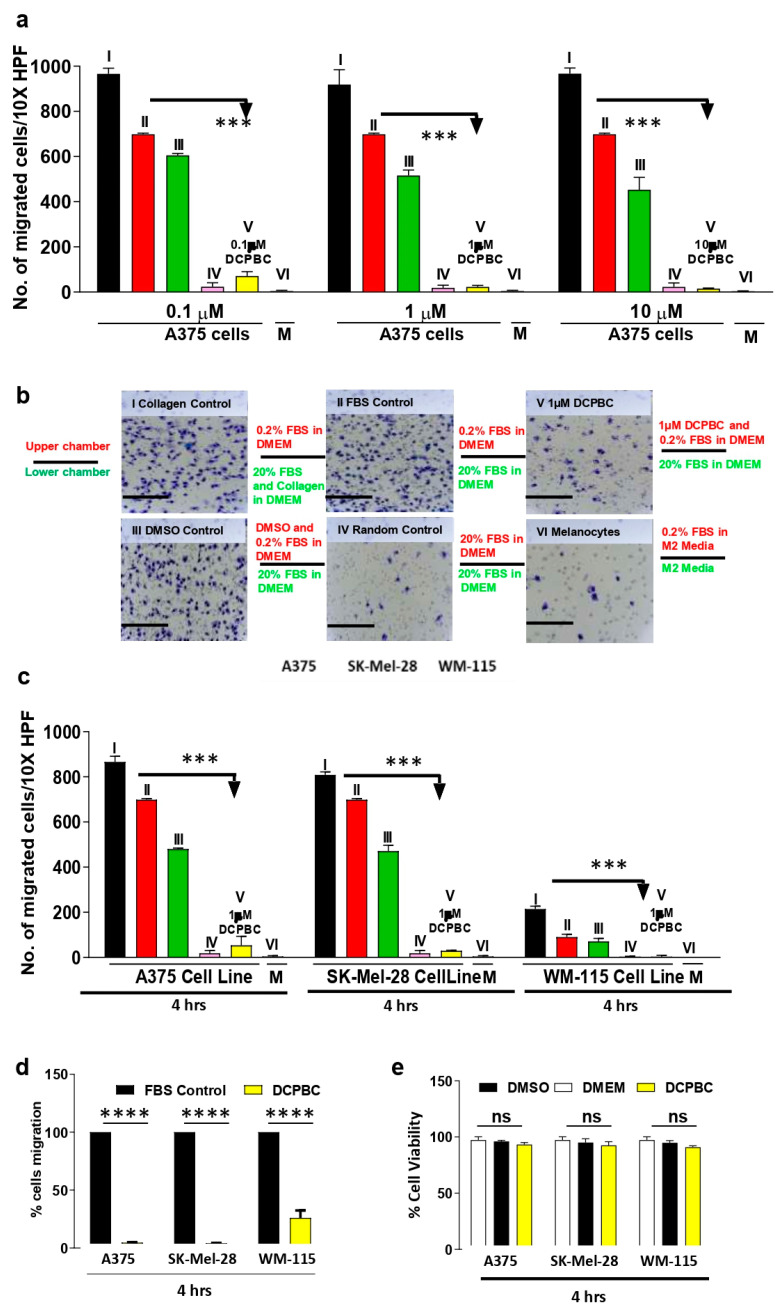
3, 3′- (3, 5-DCPBC) profoundly inhibits directed melanoma cell migration. (**a**) Migration assays were performed as detailed in materials and methods. The migration of A375 melanoma cells was tested in 3, 3′- (3, 5-DCPBC) at the indicated concentrations. Human placental type IV collagen at a concentration of 100 μg/µL dissolved in 600 µL 20% FBS in DMEM served as strong chemoattractants. The migration period was 4 h. DMEM, Dulbecco’s Modified Eagle’s Medium; DMSO; Dimethyl sulfoxide; FBS, fetal bovine serum. (I) Positive control (Collagen type IV in 20% FBS in DMEM added in the lower chamber), (II) FBS control (20% FBS in DMEM added in the lower chamber), (III) DMSO Control (DMSO in 0.2% FBS added in the upper chamber), (IV) Random Control (20% FBS in DMEM added both in the upper and lower chamber), (V) 3, 3′- (3, 5-DCPBC) at the indicated concentrations dissolved in 0.2% FBS and added in the upper chambers, (VI) Melanocytes were tested for their migratory response towards 20% FBS dissolved in M2 Media (chemoattractant). (**b**) Representative photomicrographs of the bottom side of the perforated membranes from the experiments described in Figure 2a; bars, 10 µm. (**c**) Comparison of the inhibitory activity of 1 µM 3, 3′- (3, 5-DCPBC) on the directed migration of different primary and metastatic melanoma cell lines and melanocytes. The differently tested groups (I-VI) are as described in Figure 2a. (**d**) Percentage of inhibition of the directed migration of metastatic and non-metastatic melanoma cells by 1 µM 3, 3′- (3, 5-DCPBC). (**e**) Effect of 1 µM 3, 3′- (3, 5-DCPBC) during the migration time of 4 h on the viability of non-metastatic and metastatic melanoma cells lines. Data are shown as mean ± S.D; *n* = 3 replicates; graphs represent one of three independent experiments; *** *p* < 0.0005 calculated by one-way ANOVA between 3, 3′- (3, 5-DCPBC) treated melanoma cells and controls. Bars represent the means of triplicate determinations, and error bars indicate the standard deviations.

**Figure 3 molecules-27-01172-f003:**
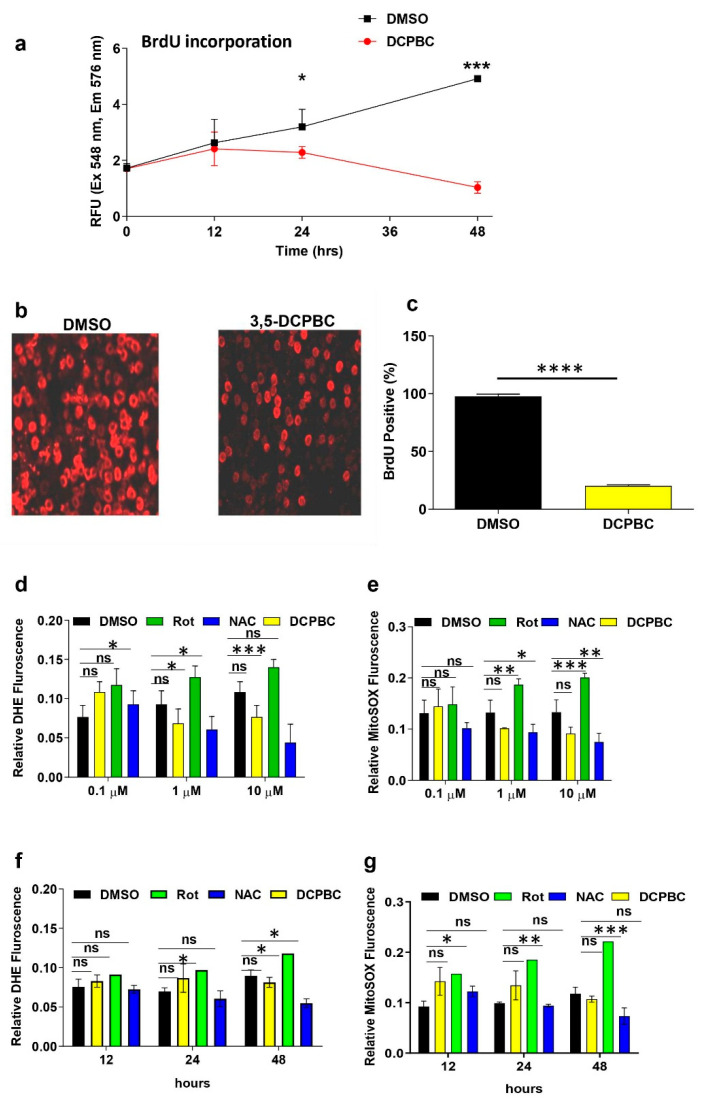
3, 3′- (3, 5-DCPBC) inhibits melanoma cell proliferation and lacks free radical scavenging properties. (**a**) A375 metastatic melanoma cells were cultured in the presence of 1 µM 3, 3′- (3, 5-DCPBC), or 1 µM Dimethyl sulfoxide (DMSO). A pulse of BrdU was provided for 2 h as detailed in material and methods. The Proliferation of BrdU incorporating into A375 melanoma cells was assessed at the indicated time points by a fluorescence spectrometer at excitation of 548 nm and emission of 576 nm, respectively. The red graph corresponds to 3, 3′- (3, 5-DCPBC) treated and the black graph to non-treated A375 melanoma cells. Data are presented as mean ± standard deviation (S. D). * *p* < 0.05, between treated and non-treated cells at 24 h. *** *p* < 0.0001 between treated and non-treated cells at 48 h. Independent experiments were repeated three times. DMSO, vehicle control; DMSO, Dimethyl sulfoxide. Asterisks indicate statistical differences determined by unpaired student’s *t*-test. (**b**) Fluorescence images of BrdU incorporation indicative A375 melanoma cell proliferation after treatment with compound 3, 3′- (3, 5-DCPBC) (right panel) or, as control with DMSO (left panel) at concentrations of 1 µM for 48 h. (**c**) BrdU positive A375 melanoma cells were quantified with original data from Figure 3b. Data are presented as mean ± S. D. *n* > = 3, *** *p* < 0.0003. Asterisks indicate the statistical difference determined by unpaired student’s *t*-test. (**d**) Dose-response relationship between 3, 3′- (3, 5-DCPBC) at the indicated concentrations and changes in cytosolic ROS levels. *N*-acetylcysteine (NAC) and rotenone (Rot) were used as ROS quenching and ROS enhancing agents (positive and negative control). DMSO served as vehicle control. (**e**) The MitoSOX assay was employed to assess dose-dependent changes in mitochondrial ROS levels, particularly mitochondrialO_2_^−^ upon treatment of A375 melanoma cells with 3, 3′- (3, 5-DCPBC) at the indicated concentrations. (**f**) Time kinetic of cytosolic ROS levels upon treatment of A375 melanoma cells with 1 µM 3, 3′- (3, 5-DCPBC). (**g**) Time kinetic of mitochondrial ROS levels upon treatment of A375 melanoma cells with 1 µM 3, 3′- (3, 5-DCPBC). (**d**–**g**); histograms showing the mean ± S. D value of relative DHE or MitoSOX fluorescence intensities indicative of distinct ROS levels. Statistical analysis was carried out by performing one-way ANOVA. Data are shown as mean ± S.D; *n* = 3 replicates; graphs represent one of three independent experiments; *** *p* < 0.0001, ** *p* < 0.001, and * *p* < 0.01 calculated by one-way ANOVA between 3, 3′- (3, 5-DCPBC) treated melanoma cells and controls. Bars represent the means of triplicate determinations, and error bars indicate the standard deviations.

**Figure 4 molecules-27-01172-f004:**
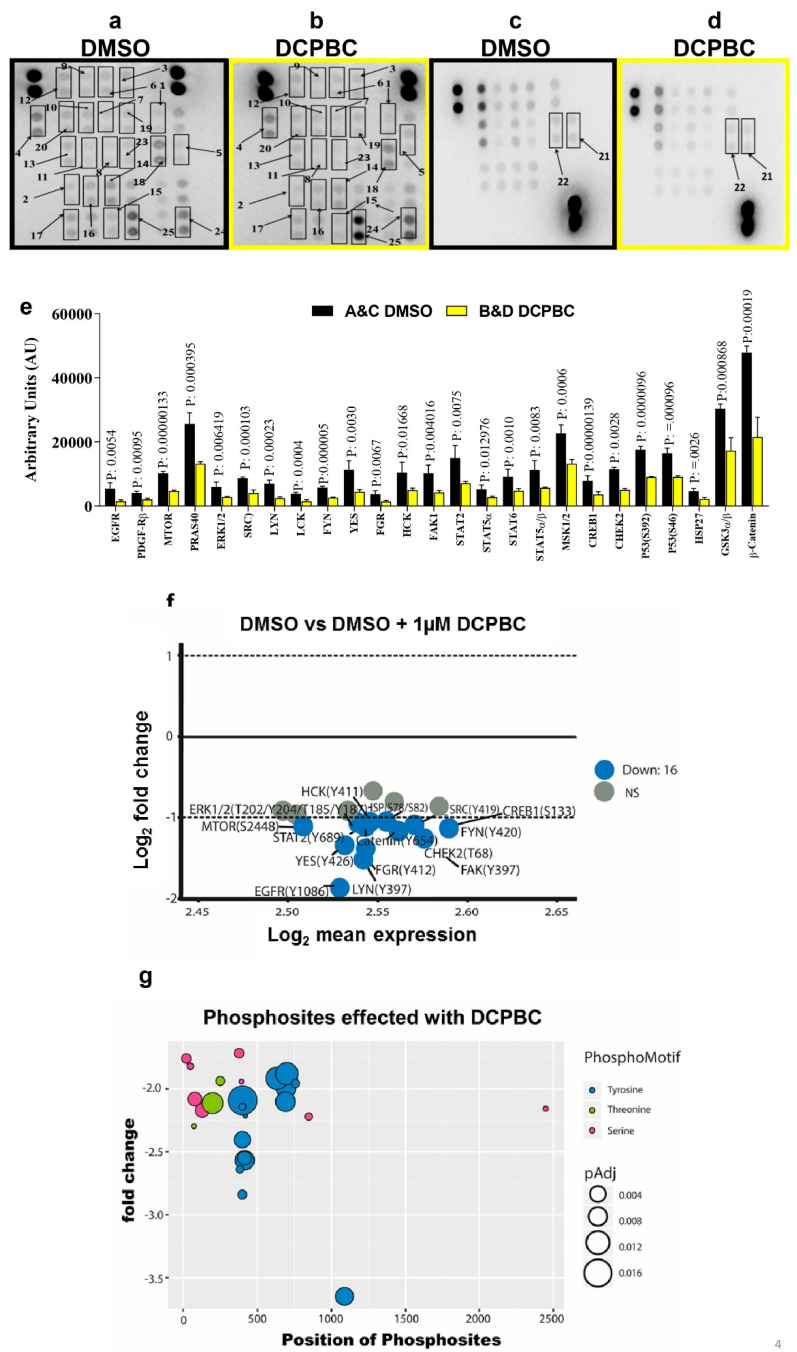
3, 3′-(3, 5-DCPBC) impacts the phosphorylation of multiple kinases. (**a**,**c**) Cell lysates from A375 melanoma cells treated with either 1 µM DMSO-treated control or (**b**,**d**) with 1 µM 3, 3′- (3, 5-DCPBC) for 4 h and thereafter were incubated with PVDF-membranes with anchored antibodies for phosphor-Y1086-EGFR (spot 1), phospho-Y-751-PDGF-β (spot 2), phosphor-S-2448-mTOR (spot 3), phosphor-T246-PRAS40 (spot 4), phospho-T202/Y204/T185/Y187--ERK1/2 (spot 5), phosphor-Y-419-Src (spot 6), phospho-Y-397- Lyn (spot 7), phosphor-Y-394-Lck (spot 8), phosphor-Y-420-Fyn (spot 9), phosphor-Y426-Yes (spot 10), phosphor-Y-412-Fgr (spot 11), phosphor-Y-411-Hck (spot 12), phosphor-Y-397-FAK (spot 13), phosphor-Y689-STAT2 (spot 14), phosphor-Y-694- STAT5α (spot 15), phosphor-Y641-STAT6 (spot 16), phosphor-Y-694/699-STAT5α/β (spot 17), phosphor-S376/S360-MSK1/2 (spot 18), phosphor-S133-CREB (spot 19), phosphor-T68-Chk-2 (spot 20), phosphor-S-392-p53 (spot 21) and phosphor-S-46-p53 (spot 22), phosphor-S78/S82-HSP27 (spot 23), phosphor-S21/S9-GSKα/β (spot 24), phosphor-β-catenin (spot 25). Membranes were developed with appropriate secondary antibodies, and the spots were detected using a chemiluminescence-based assay as detailed in material and methods. (**e**) Densitometric analysis for the phosphorylation state of multiple kinases from Figure 4a–d employing gel quant software. Results are expressed as “Arbitrary Density units” and presented as mean ± S.D. Data are analyzed by performing one-way ANOVA, and the statistical difference is shown as * *p* values. (**f**) Differential Phospho-regulation is depicted by the MA plot. The *x*-axis represents the mean expression, and the *y*-axis represents log2 fold change. The genes that are more than two-fold differentially regulated are shown in blue, whereas the non-significant one is shown in grey. (**g**) The phosphorylation motifs were analyzed from the 16 differentially downregulated phosphokinase motifs, as shown in Figure 4e. Out of the significantly downregulated phosphor kinase motifs by 3, 3′- (3, 5-DCPBC), tyrosine kinase was the most regulated, as represented in blue. The serine and threonine phosphor motifs are shown in pink and green colors. The *x*-axis represents the amino acid position number of the phosphor-sites in the reference protein sequence. Tyrosine kinase Phospho-sites were majorly affected with 3, 3′- (3, 5-DCPBC) treatment.

**Figure 5 molecules-27-01172-f005:**
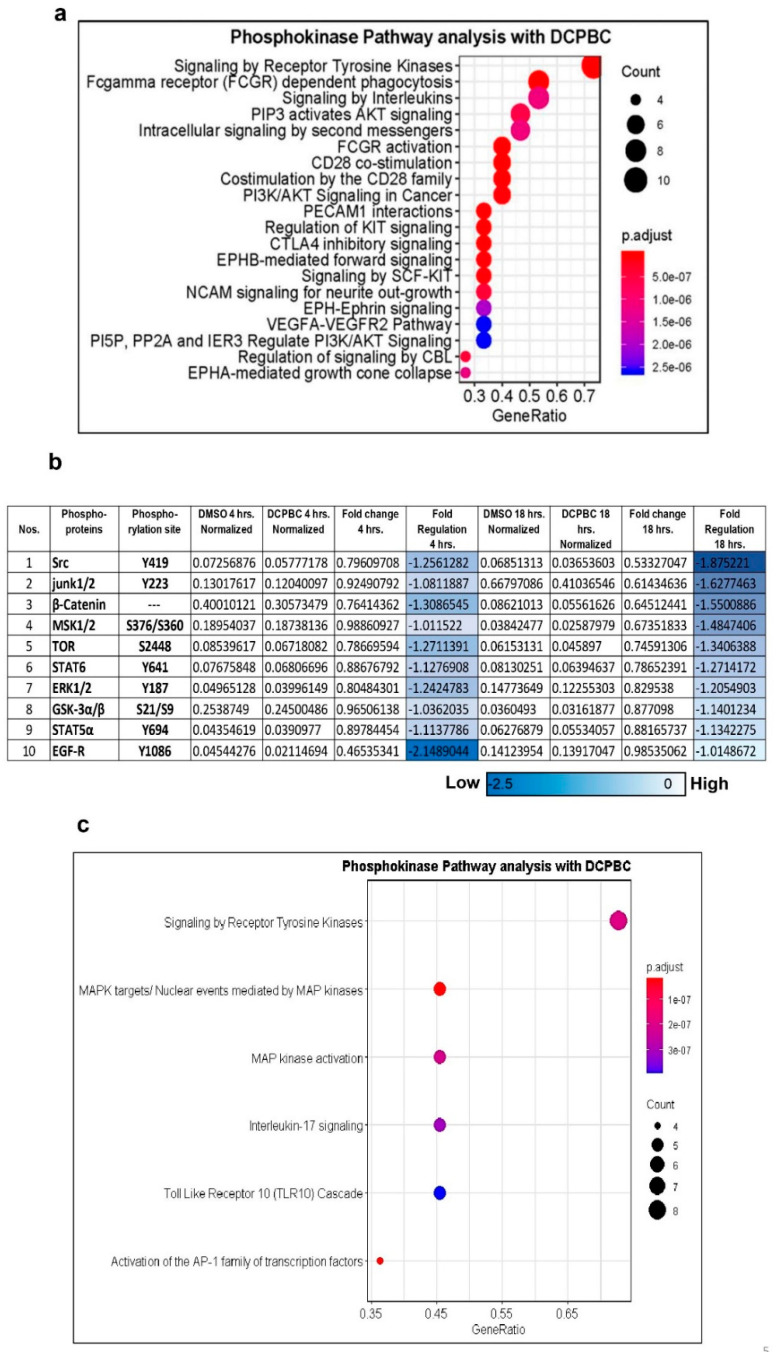
3, 3′- (3, 5-DCPBC) reduces the phosphorylation (activation) of multiple kinases in melanoma cells. (**a**) represents pathway analysis carried out using gene names from these 16 regulated phosphor-sites. The *x*-axis represents the gene ratio, whereas the associated pathways are represented in the *y*-axis. The color of the circles represents the level of significance as *p*-value Adjusted. (**b**) A375 metastatic melanoma cells were treated with 1 µM 3, 3′- (3, 5-DCPBC) for 18 h as described in Figure 4. Densitometry data were obtained using gel quant software and normalized for 4 and 18 h with β-actin. Results are expressed as “Arbitrary Density units” and presented a mean ± S.D. for 3 independent experiments. Data were analyzed by performing Two-way ANOVA, and the statistical difference is shown as * *p* values. The table shows a list of selective kinases whose phosphorylation sites were downregulated at 4 and 18 h. Dark blue represents strong suppression of the kinase activity, lighter blue indicates lesser suppression. (**c**) In silico pathway analysis of significantly down-regulated genes between 4 vs. 18 h with 3, 3′- (3, 5-DCPBC) treatment. The pathways are represented on the *y*-axis and the Gene ratio on the *x*-axis. Color represents the level of significance *p*-Value adjusted. Data is analyzed by one-way ANOVA with time and gene expression as individual factors, and the statistical difference is shown as * *p* < 0.05, ** *p* < 0.05, and *** *p* < 0.001 when compared between treated and control cells.

**Figure 6 molecules-27-01172-f006:**
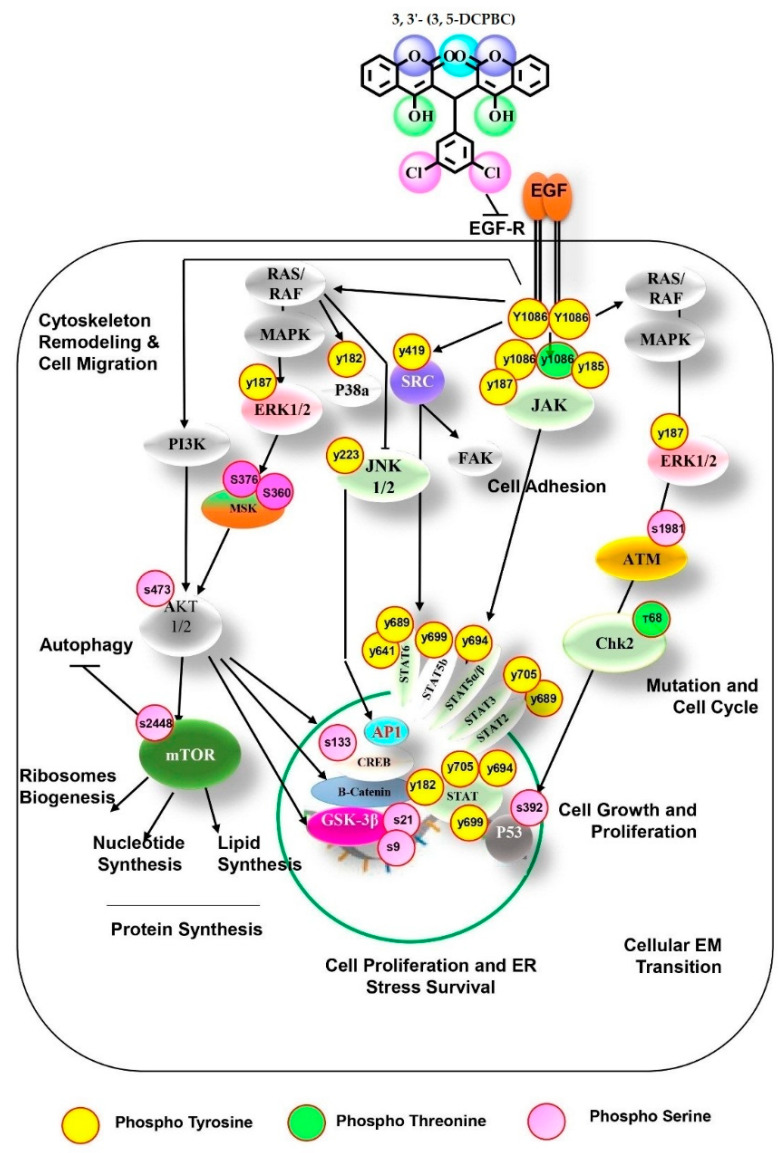
3, 3′- (3, 5-DCPBC) impacts important regulatory hubs in melanoma progression. Summary scheme depicting 3, 3′- (3, 5-DCPBC) targeting EGFR and SRC kinases, AKT/mTOR, and RAS/RAF/MAP kinases. Taken together, 3, 3′- (3, 5-DCPBC) shows excellent tyrosine kinase and serine kinase inhibiting properties. Most excitingly, it is endowed with the potential to target multiple kinases to prevent cellular migration (via SRC), proliferation, and protein synthesis (via AKT/mTOR). The Red circular outline indicates the post-translationally regulated phosphokinase sites which were found to be significantly down-regulated in melanoma cells after 3, 3′- (3, 5-DCPBC) treatment compared to controls. The ash-colored circular nodes represent intermediate signaling pathway transcription factors whose phosphorylation site was not changed. They are mentioned to depict the complete pathways. The yellow, green, and pink phosphorylation sites represent tyrosine, threonine, and serine, respectively.

## Data Availability

All data generated or analyzed on 3, 3′- (3, 5-DCPBC) during this study are included in this article.

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
