# Peer review of "3, 3′- (3, 5-DCPBC) Down-Regulates Multiple Phosphokinase Dependent Signal Transduction Pathways in Malignant Melanoma Cells through Specific Diminution of EGFRY1086 Phosphorylation"

_molecules, 2022, doi:10.3390/molecules27041172_

Round 1
Reviewer 1 Report
The manuscript entitled “3’ 5 DCPBC; A Novel Bis-Coumarin Down-regulates Multiple Phos-phokinase Dependent Signal Transductions Pathways in Malignant Melanoma Cells through Specific Diminution of EGFRY1086 Phosphory-lation” describes the molecular mechanism of the action high active 3,5-DCPBC as anti-melanoma agent. In addition to a deep biochemical study of the action, some major corrections should be done.
- Article title contains orthographic mistakes and should be corrected (3,5-DCPBC, Phosphokinase, Phosphorylation; marked by the yellow color).
- 3,5-DCPBC is not a new compound! 3,5-DCPBC firstly was described in ‘Molecules 2015, 20, 17614–17626’. Authors should correct all parts of the manuscript (marked by the yellow color).
- 3,5-DCPBC was previously studied as an anti-cancer agent (doi:10.3390/molecules200917614). This information should be added to the introduction part.
- The name of 3,5-DCPBC should be corrected to 3,3'-((3,5-dichlorophenyl)methylene)bis(4-hydroxy-2H-chromen-2-one) (P. 2 line 68 and Figure 1)
- There is no positive control for no one famous FDA-approved drugs for melanoma treatment in determining cytotoxicity.
- Authors should correct all references according to the rules of the Journal (marked by the yellow color) https://www.mdpi.com/journal/molecules/instructions#references

Author Response
Response to Reviewer 1 Comments
Point 1: Article title contains orthographic mistakes and should be corrected (3,5-DCPBC, Phosphokinase, Phosphorylation; marked by the yellow color).
Response 1: Article title has been corrected as follows and highlighted with ‘yellow color
3, 3'- (3, 5-DCPBC): A Novel Bis-coumarin Down-regulates Multiple Phosphokinase Dependent Signal Transduction Pathways in Malignant Melanoma Cells through Specific Diminution of EGFRY1086 Phosphorylation
Point 2: 3,5-DCPBC is not a new compound! 3,5-DCPBC firstly was described in ‘Molecules 2015, 20, 17614–17626’. Authors should correct all parts of the manuscript (marked by the yellow color).
Response 2: 3, 3'- (3, 5-DCPBC) is indeed a new compound/ a novel derivative of previously reported bis-coumarin class. Bis -coumarin is a class of compound that has been previously reported in literature. Due to its tremendous medicinal/biological importance a number of bis-coumarin derivatives have already been synthesized and published in literature, and are still synthesizing and reporting. When a derivative of a previously existing class synthesized it is recognised as a new if it is having unique chemical and physical properties, unique molecular weight, unique structure, unique functional groups then all other previously existing derivatives. Due to having differences in all above characteristics, a derivative recognises with its own IUPAC name.Our collaborator synthesized and reported a series of new bis-coumarin derivatives of previously existing bis- coumarin class during 2019.
https://doi.org/10.1016/j.bioorg.2019.103170
and sufficiently cited in your manuscript as reference 30 in the introduction and results section 2.1.
These all derivatives are unique/ different in their structure, physiochemical properties, molecular weight, among all previously reported derivatives of bis- coumarin class, and never been tested against melanoma or any other cancerous cell lines. 3, 3'- (3, 5-DCPBC) is one derivative from them designated as compound 22 of the series in above mention research article with the IUPAC name in the section 4.24.
3, 3′-((3′′,5′′-Dichlorophenyl)methylene)bis(4-hydroxy-2Hchromen-2-one) (22)
None of the derivative with the exact above name is described in ‘Molecules 2015, 20, 17614–17626’doi: 10.3390/molecules200917614
OR
Anywhere in literature.
For more clarification kindly search the compound/ Derivative with its IUPAC name.
Point 3. 3, 5-DCPBC was previously studied as an anti-cancer agent (doi:10.3390/molecules200917614). This information should be added to the introduction part.
Response 3: 3, 3'- (3, 5-DCPBC) has not been tested against any cancerous cell lines specifically any melanoma cell lines. It has been newly synthesized by our collaborator during 2019 ()
Point 4: The name of 3, 5-DCPBC should be corrected to 3,3'-((3,5-dichlorophenyl)methylene)bis(4-hydroxy-2H-chromen-2-one) (P. 2 line 68 and Figure 1)
Response 4: The full and short name of the compound has been corrected and highlighted with yellow in the (P. 2 line 68 and Figure 1) tittle and everywhere.
Full name [3, 3' - ((3", 5"-Dichlorophenyl) methylene) bis (4-hydroxy-2H-chromen-2-one)] [3, 3'- (3, 5-DCPBC)]
Short name/ Abbreviation 3: 3, 3'- (3, 5-DCPBC)
Point 5: There is no positive control for no one famous FDA-approved drugs for melanoma treatment in determining cytotoxicity.
Response 5: Our goal was to look for small molecule inhibitors which could act over a broad spectrum of Melanomas and not be restricted to BRAF mutated or non-mutated melanomas against which clinical standards are available, as well as at the same time be non-toxic to Melanocytes, as many of the DNA intercalating clinical standards are toxic to melanocytes.
If the panel of small molecule inhibitors would have been screened against any particular clinical standards, it might not give us similar results if another clinical standard or another melanoma type was compared. Hence for the initial screening purpose, we used the vehicle (DMSO) as a standard.
Point 6: Authors should correct all references according to the rules of the Journal (marked by the yellow color) https://www.mdpi.com/journal/molecules/instructions#references
Response 6: All the references has been corrected according to the rules of the Journal and marked with yellow color.

Reviewer 2 Report
Review comments of manuscript Molecules-1496558 entitled:
3’ 5 DCPBC; A Novel Bis-Coumarin Down-regulates Multiple Phos-phokinase Dependent Signal Transductions Pathways in Malignant Melanoma Cells through Specific Diminution of EGFRY1086 Phosphory-lation
In this manuscript, the author investigated the effect of the new bis-coumarin derivative [3, 3’ - ((3”, 5”-Dichlorophenyl) methylene) bis (4-hydroxy-2H-chromen-2-one)] (3, 5-DCPBC) on human melanoma cell survival, growth, prolifer- ation, migration, intracellular redox state, and deciphered associated signaling pathways. The study in the paper could lay a good foundation for an efficient alternative approach to currently established therapies. I think there are some issues to be addressed before further review on the result of this manuscript, and I would like to address several points as follows:
Major comments:
- In Fig1, Fig1b has the same name as Fig1C.
In Fig2, the bars of the ruler should be marked at the bottom right of Fig2b, and the name of the y-axis “%cell migration” of Fig2e should be checked.
In Fig2e, the cytotoxicity of 3, 5-DCPBC was only investigated on A375 cells, and the data on SK-Mel-28 cells and WM-115 cells should be added.
- The statistic descriptions in the article are inconsistent, such as mean ± SEM on page 4, line 105, and mean ± S.D on page 6, line 167. There are many such inconsistencies in the text.
- On page 9, Fig4f and Fig4g are too obscure to recognize.
- On page11, Fig5, lines 327 to 328, “Data is analyzed by two-way ANOVA and the statistical difference are shown as *p values,” however, there is no relevant statistical test in the legend of Fig3.
- On page 14, item 4.9, line 477, “R studio was used for data analysis and networking analysis respectively,” 5he version number of the software should be marked here. Besides, the pathway enrichment method and networking analysis in the text (e.g., Fig4 and Fig5) should be described in detail.
Minor comments:
- On page 2, item 2.1, the alphabetical numbering of the figure is not uniform in case, and such problems also exist in other paragraphs and figure legends.
- On page 6, line 153, “600 l 20%FBS” should be checked. Besides, it seems that the sentence of lines 182 to 184 is missing punctuation.
- On page 13, item 4.4, line 410, line 412, the superscript or subscript of the number should be checked, and there are such errors elsewhere in the text.

Author Response
Response to Reviewer 2 Comments
Point 1: In Fig1, Fig1b has the same name as Fig1C.
Response 1: Fig 1, Fig1b and Fig 1c has been corrected
Point 2: In Fig2, the bars of the ruler should be marked at the bottom right of Fig2b, and the name of the y-axis “%cell migration” of Fig2e should be checked.
In Fig2e, the cytotoxicity of 3, 5-DCPBC was only investigated on A375 cells, and the data on SK-Mel-28 cells and WM-115 cells should be added.
Response 2: In Fig2, the bars of the ruler should have been shifted to the bottom right of Fig2b, and the name of the y-axis of Fig2e has been corrected as “%cell viability”
In Fig2e, the data of cytotoxicity of 3, 3'- (3, 5-DCPBC) on the SK-Mel-28 cells and WM-115 cells has been added.
Point 3. On page 9, Fig4f and Fig4g are too obscure to recognize.
Response 3: On page 9, clear good quality of Fig4f and Fig4g has been added
Point 4: On page11, Fig5, lines 327 to 328, “Data is analyzed by two-way ANOVA and the statistical difference are shown as *p values,” however, there is no relevant statistical test in the legend of Fig3.
Response 4: On page11, Fig5, lines 327 to 328, there was a typing error that has been been corrected and relevant statistical test in the legend of Fig3 has ben added.
Point 5: On page 14, item 4.9, line 477, “R studio was used for data analysis and networking analysis respectively,” 5he version number of the software should be marked here. Besides, the pathway enrichment method and networking analysis in the text (e.g., Fig4 and Fig5) should be described in detail
Response 5: On page 14, item 4.9, line 477, “R studio version number of the software has been added. (e.g., Fig4 and Fig5) has been described in detail Besides, the pathway enrichment method and networking analysis in the text.
Minor comments:
Point 1: On page 2, item 2.1, the alphabetical numbering of the figure is not uniform in case, and such problems also exist in other paragraphs and figure legends.
Response 1: On page 2, item 2.1, and other paragraphs and figure legends. the alphabetical numbering of the figures has been corrected.
Point 2: On page 6, line 153, “600 l 20%FBS” should be checked. Besides, it seems that the sentence of lines 182 to 184 is missing punctuation..
Response 2: On page 6, line 153, “600 l 20%FBS” has been corrected as µl the sentence of lines 182 to 184 has been corrected and highlighted with yellow.
Point 3: On page 13, item 4.4, line 410, line 412, the superscript or subscript of the number should be checked, and there are such errors elsewhere in the text.
Response 3: The superscript or subscript of the On page 13, item 4.4, line 410, line 412, and everywhere in the has been corrected and highlighted with yellow color.

Round 2
Reviewer 1 Report
Dear Authors of the manuscript. After reading your quick response to comments, I think that some points need to be discussed. In addition, I believe that this manuscript needs profound corrections.
- In a previous review of this manuscript was indicated that the compound under study is already known and was first synthesized (compound #2) in “Molecules. 2015, 20, 17614–17626 ”(https://doi.org/10.3390/molecules200917614). Therefore, the claim that this compound was first synthesized in the article “Bioorganic Chemistry. 2019, 91, 103170” (https://doi.org/10.1016/j.bioorg.2019.103170) is not true. To determine if a compound is new, authors should use specialized chemical structure database (Scifinder or Reaxys) using the exact chemical structure.
- Also, in the article mentioned above, the authors investigated the antibacterial and antitumor activity for the compound #2. This information should be added to the Introduction. Moreover, the authors should correct all other parts of the manuscript, as the compound is not new.
- To better understand the significance of the obtained results, the authors should add positive control in the study of cytotoxicity.
- Authors should carefully check the Reference list and correct it.
Author Response
Response to comment 1
We have checked the claim of novelty of compound by using specialized chemical structure database (Scifinder) and have found that both compounds (compound #2 presented in “Molecules. 2015, 20, 17614–17626 ”(https://doi.org/10.3390/molecules200917614) and our studied compound 3, 3'- (3, 5-DCPBC) whose synthesis was first published in “Bioorganic Chemistry. 2019, 91, 103170” (https://doi.org/10.1016/j.bioorg.2019.103170) are same, therefore we have removed word new from the text and from the tittle of manuscript.
Response to comment 2
We have added in the Introduction the antibacterial and antitumor activity for the compound #2. Moreover, we have corrected all other parts of the manuscript, as the compound is not new.
Response to comment 3
New analysis with the positive control cannot be done within the given time, hence we would request publish current results.
Response to comment 4
We have carefully check and corrected the Reference list.